# Brain-Derived Neurotrophic Factor (BDNF) as a Marker of Physical Exercise or Activity Effectiveness in Fatigue, Pain, Depression, and Sleep Disturbances: A Scoping Review

**DOI:** 10.3390/biomedicines13020332

**Published:** 2025-01-31

**Authors:** Nada Lukkahatai, Irvin L. Ong, Chitchanok Benjasirisan, Leorey N. Saligan

**Affiliations:** 1School of Nursing, Johns Hopkins University, Baltimore, MD 21205, USA; cbenjas1@jhmi.edu; 2Research Development and Innovation Center, Our Lady of Fatima University, Valenzuela City 1440, Philippines; irlong@fatima.edu.ph; 3Department of Nursing and Health Sciences, Elmhurst University, Elmhurst, IL 60126, USA; 4National Institute of Nursing Research, National Institutes of Health, Bethesda, MD 20892, USA; saliganl@mail.nih.gov

**Keywords:** brain-derived neurotrophic factor, human BDNF protein, exercise, symptom management

## Abstract

**Background/Objectives**: Brain-derived neurotrophic factor (BDNF) has been investigated as a potential mechanistic marker or therapeutic target to manage symptoms such as fatigue, pain, depression, and sleep disturbances. However, the variability in BDNF response to exercise or physical activity (exercise/PA) and its clinical relevance in symptom management remains unclear. This scoping review assesses existing studies exploring the relationships between exercise/PA, symptoms, and BDNF levels, specifically focusing on fatigue, pain, depression, and sleep disturbances in adults. **Methods**: Relevant studies indexed in PubMed and CINAHL were identified. Using systematic review software, two reviewers independently screened and evaluated full texts, based on the following criteria: human studies reporting BDNF levels in adults, using exercise/PA interventions, assessing symptoms (pain, fatigue, depression, and/or sleep disturbance) as outcomes, and published in English. **Results**: Of 950 records, 35 records met the inclusion criteria. While exercise/PA is broadly supported for managing symptoms, 74.3% (n = 26) of studies reported increased BDNF levels, and only 40% (n = 14) showed significant increases following exercise/PA. Only 14% (n = 5) of studies demonstrated a significant relationship between changes in BDNF and symptoms. No significant differences in BDNF levels and symptoms were observed between different types of exercise (e.g., aerobic vs. strength vs. flexibility/stretching) and PA. **Conclusions**: The current literature provides insufficient evidence to confirm BDNF as a marker for exercise/PA effectiveness on symptoms. Further clinical investigations are needed to validate its potential as a therapeutic target.

## 1. Introduction

Brain-derived neurotrophic factor (BDNF) is one of the neurotrophin proteins that plays a central role in neurogenesis, synaptogenesis, and synaptic plasticity [1,2]. Coded by the *BDNF* gene, this neurotrophin is highly expressed in the brain glutamatergic neurons, glial cells, and microglia [2]. The key regions where BDNF is predominantly concentrated include the hippocampus, cortex, and basal forebrain [2,3]. The hippocampus is essential for learning and memory processes, with BDNF playing a crucial role in synaptic plasticity and the survival of neurons within this area. The cortex, responsible for higher cognitive functions, and the basal forebrain, which modulates attention and arousal, exhibit significant BDNF expression. Additionally, BDNF is present in the amygdala, a region involved in emotional responses. The widespread expression of BDNF across these regions suggested its importance in neuroplasticity and its potential as a therapeutic target in various neurological and psychiatric conditions.

The mature isoform of BDNF (mBDNF) binds to the tropomycin receptor kinase B (TrkB) to regulate numerous brain physiological processes, such as the apoptosis and survival of neurons, and learning- and memory-process-dependent synaptic plasticity, which are vital for brain development and function [4,5]. Evidence suggests a potential connection between BDNF activity and sigma-1 receptors, indicating that sigma-1 receptors may enhance BDNF signaling efficiency and neuroprotective effects, contributing to improved neuronal resilience and functional recovery in neurodegenerative and psychiatric conditions. Sigma-1 receptors regulate key cellular processes such as excitotoxicity, oxidative stress, ER stress, and mitochondrial dysfunction [6]. Their agonists have shown neuroprotective properties by promoting cell survival, stabilizing mitochondrial function, and reducing neuroinflammation—factors associated with symptoms like fatigue, pain, depression, and sleep disturbances [7,8]. Reduced BDNF levels, particularly in the hippocampus and prefrontal cortex, have been observed in individuals with depression and cognitive impairment, suggesting its potential role in managing these symptoms.

The relationship between BDNF levels in the brain and circulating BDNF in the bloodstream remains complex and not yet fully understood. While some studies suggest that circulating BDNF may reflect brain BDNF activity [9], others highlight the significant contribution of peripheral sources, such as platelets and skeletal muscle, to circulating levels [10,11]. Circulating BDNF can be detected in endothelial cells, cardiomyocytes, smooth muscle cells [12], and blood cells such as leukocytes and platelets [13], which indicates its potential role in various disease processes, including neurodegenerative disorders [14,15], cardiovascular diseases [16,17], and cancer [18]. The Enzyme-Linked Immunosorbent Assay (ELISA) for serum or plasma samples, Western blotting for tissue protein detection, and Quantitative PCR (qPCR) for gene expression measurement are commonly used methods for measuring circulating BDNF [19]. The range of BDNF levels in circulation varies considerably depending on the individual and health conditions, with typical serum BDNF concentrations ranging from 20 to 30 ng/mL in healthy adults [20]. Higher BDNF levels are often associated with improved cognitive function, while lower levels are linked to conditions such as depression and neurodegenerative diseases [1].

There is growing interest in BDNF as a potential biomarker for various chronic condition-related symptoms. Studies have shown the possible roles of BDNF, both at the gene and protein levels, in chronic condition-related symptom experiences such as fatigue [21,22,23], chronic pain [24,25,26], depression [27,28], sleep disturbance [29,30], and memory impairment [1,31]. For instance, low serum BDNF levels have been associated with worsening fatigue during external beam radiation therapy in cancer patients [22] and adults with osteoarthritis [32]. Similarly, individuals with depression had lower serum BDNF levels compared to healthy volunteers [29,33]. With the focus on improving cognitive function, current evidence supports the role of BDNF as a potential underlying mechanism to explain physical exercise benefits [34,35]. These findings suggest that BDNF could be a useful biomarker in understanding symptom experiences across multiple chronic conditions [36]. Guided by the American College of Sports Medicine (ACSM) Physical Activity Guidelines for Americans, exercises were classified into aerobic, strength, and stretching/flexibility based on their distinct physiological effects [37]. Aerobic exercises, such as running, cycling, and swimming, elevate heart rate and improve cardiovascular and respiratory function. Strength training exercises, including weightlifting and bodyweight exercises, focus on building muscle strength and improving metabolic function. Flexibility and stretching exercises, including yoga and various stretching routines, aim to improve joint range of motion, reduce stress, and promote recovery. These categories highlight the varied impacts of physical activity on BDNF levels and symptom management, particularly in chronic conditions, to better understand their contributions to improving symptoms such as fatigue, pain, and depression.

Several systematic reviews and meta-analyses examine the impact of exercise or physical activity (exercise/PA) on BDNF levels and associated clinical outcomes, particularly in age-related neurodegenerative populations and older adults [36,38,39]. Only one review focuses explicitly on symptoms, highlighting the connection between resistance training, increases in BDNF, and reductions in depressive symptoms [36]. This review identifies the variability in effectiveness due to differing health conditions. A significant gap remains in understanding how exercise-induced BDNF changes might help alleviate specific symptoms, including fatigue, pain, sleep disturbances, and depression. Additionally, studies suggest variations in BDNF response depend on factors such as type, duration, and intensity of physical exercise and activity, making it challenging to generalize findings across different populations. This challenge is particularly evident in younger individuals and those with chronic conditions [14,40,41]. These complexities highlight the need for further research to explore BDNF’s role across various symptoms and diverse chronic conditions.

This scoping review explores the potential role of BDNF as a biomarker in exercise-based intervention for fatigue, pain, depression, and sleep disturbance management. We focus on the symptoms of fatigue, pain, depression, and sleep disturbances, which are among the most prevalent and impactful symptoms reported in individuals with chronic conditions. These symptoms are closely linked, often exacerbating each other, and are central to many intervention strategies to improve the quality of life in affected populations. In this review, the term “depression” is used to refer to a general emotional state characterized by feelings of sadness or low mood, which may occur in response to a particular life event. “Clinical depression” (or Major Depressive Disorder) is a more severe, diagnosable condition characterized by persistent symptoms such as profound sadness, loss of interest in usual activities, and functional impairment that lasts for at least two weeks and interferes with daily life. Similarly, “anxiety” refers to mild emotional reactions to stress involving feelings of worry or apprehension. In contrast, “anxiety disorders” refer to a group of mental health conditions characterized by excessive, persistent anxiety or fear that is disproportionate to the situation and disrupts daily functioning. Examples of anxiety disorders include Generalized Anxiety Disorder (GAD), panic disorder, and social anxiety disorder.

It seeks to identify key findings on BDNF as a marker for assessing the effectiveness of physical exercise and activity in managing symptoms and examines the association between exercise-/PA-induced changes in BDNF and the alleviation of specific symptoms, such as fatigue, pain, sleep disturbances, and depression. By focusing on these connections, this review offers insights into how BDNF may serve as a valuable biomarker in exercise-/PA-based interventions for fatigue, pain, depression, and sleep disturbance management across various conditions.

## 2. Methods

In this manuscript, the terms “exercise” and “physical activity (PA)” are used interchangeably to refer to any form of physical movement that enhances or maintains physical health.

### 2.1. Literature Search

The scoping review was conducted using a guide by Munn et al. [42], an approach by Sucharew and Sucharew et al. [43], and the framework by Arksey and O’Malley et al. [44], and the Preferred Reporting Items for Systematic Reviews and Meta-Analyses (PRISMA) extension for scoping reviews [45]. Databases including PubMed and Cumulative Index of Nursing and Allied Health Literature (CINAHL) were searched with no restrictions on publication year, using the following keywords: “exercise OR physical activity”, “Brain-Derived Neurotrophic Factor”, OR “BDNF”, “fatigue”, “pain”, “depression”, or “sleep”. Additionally, subject headings (MH “Sleep+”), (MH “Depression+”), (MH “Pain+”), (MH “Fatigue+”), (MH “Exercise+”), (MH “Physical Activity”), and (MH “Brain-Derived Neurotrophic Factor”) were used in CINAHL. MeSH (Medical Subject Headings) terms “Sleep”, “Depression”, “Pain”, “Fatigue”, “Brain-Derived Neurotrophic Factor”, and “Exercise” were used in the PubMed search (Appendix A). The automatic filter was used to include studies in “English” and “Human”. Only two databases, PubMed and CINAHL Plus, were used as they provide comprehensive coverage of biomedical and healthcare-related studies relevant to our research question. These two databases were selected based on their established relevance in biomedical research and health science. We believe they capture a broad range of relevant articles in our field. Although including additional databases could potentially increase coverage, we found these two databases sufficient for the scope of this scoping review. The most recent search was conducted in September 2024. Search results were imported from both databases to Covidence (www.covidence.org), an online tool for systematic review management (Veritas Health Innovation, Melbourne, Australia). Covidence was used to streamline the systematic review process, including study selection, data extraction, and reviewer collaboration. It functioned as a management tool to ensure consistency, enabling efficient co-working and progress tracking throughout the systematic review process. However, Covidence does not have the capability to calculate percentages or automatically extract the data from full texts.

### 2.2. Study Selection

Two reviewers (N.L., I.L.O.) independently screened the titles and abstracts. We utilized the Covidence to facilitate the screening process. Title and abstract screening and full-text reviews were conducted independently by two reviewers within the Covidence platform, with discrepancies resolved through discussion or a third reviewer if necessary. The criteria for title and abstract screening included (1) studies involving human subjects; (2) examination and reporting of BDNF levels; (3) inclusion of exercise or physical activity; (4) studies conducted with healthy volunteers or individuals with physical conditions (non-communicable chronic conditions, i.e., cancer, neurological disorder, or renal disease); and (5) publication in English. Animal studies, studies involving individuals with mental health conditions (e.g., schizophrenia, bipolar disorder, clinical depression, anxiety disorder, eating disorders, etc.), studies involving children (age younger than 18 years old), and non-English papers were excluded.

The two reviewers assessed the full-text articles that passed the title and abstract screening based on additional selection criteria: (1) studies involving human subjects; (2) participants aged 18 years or older; (3) investigation of at least one of the following symptoms—pain, fatigue, sleep disturbance, or depression (with depression defined as a general mood state)—as outcome variables; and (4) use of an exercise or physical activity program as an intervention. The exclusion criteria were (1) studies involving people with mental health conditions and (2) review papers. Critical appraisal tools for quasi-experimental studies and randomized controlled trials by the Joanna Briggs Institute (JBI) [46,47] were used to assess the studies’ methodological quality.

### 2.3. Data Extraction

A standardized charting form was developed manually to systematically extract key data from each included study. The data charting process involved recording study characteristics, including year of publication, country, and study participants. Intervention details included type (e.g., aerobic exercise, multimodal exercise), duration, frequency, and supervision. Guided by the ACSM Physical Activity Guideline for Americans [37], we classified types of activities into PA and exercise, which include three main types: aerobic exercise, strength training, and stretching/flexibility exercises. Symptom outcomes such as fatigue, pain, depression, and sleep disturbance were documented, along with whether significant improvements were reported. Details on BDNF measurements, including whether BDNF was assessed via plasma or serum, any specific timing of blood collection, and precautions taken to minimize circadian rhythm influences on BDNF levels, were also manually charted.

Additionally, we recorded studies investigating BDNF polymorphisms or genetic variations and those analyzing the relationship between changes in BDNF levels and symptom improvements. The data extraction and charting process was conducted manually by N.L. and I.L.O. and organized into a summary of evidence table format. Covidence was also used during the data extraction phase to provide structured templates that enabled both reviewers to systematically collect relevant study characteristics, interventions, and outcomes. The summary of the evidence table was then reviewed and iteratively refined by all authors to ensure comprehensive data extraction across studies.

After generating results in a summary of evidence table format, the potential roles of BDNF were identified as follows: (1) symptom-related biomarkers (linking BDNF levels with symptoms), (2) exercise/PA responses, (3) mediator, and (4) inactive (no correlation). Two reviewers (N.L. and I.L.O.) independently classified the BDNF role for each study based on the study’s aims, methods, and results. The results were compared between the two reviewers, and any discrepancies were resolved through discussion. A third reviewer (L.N.S.) was consulted to ensure an accurate classification if a consensus could not be reached.

## 3. Results

The search strategy yielded 950 records after removing 615 non-human and non-English reports, as well as three duplicates. A total of 258 records were removed during the title and abstract screening, leaving 74 reports for full-text retrieval. Two reports were unretrievable, and 35 reports were excluded based on the inclusion criteria. The most frequent reasons for exclusion during the full-text screening process were the absence of measures for BDNF symptoms, physical activity, or exercise (n = 28); studies focusing on psychological disorders or including participants under 18 years old (n = 6); and review articles (n = 3). Finally, 35 reports from 34 studies (two from the same study) met the inclusion criteria of this scoping review with acceptable methodological quality using the JBI [46,47] critical appraisal tools. The PRISMA flow (Figure 1) summarizes the screening and selection process.

More than half of the selected reports (n = 21) were published between 2019 and 2024. Notably, there was an uptick in the number of published papers on this topic in recent years. These articles were from either the Americas (43.2%), Europe (35.1%), or the Western Pacific (e.g., Australia, China, Taiwan, etc.) (21.6%). The majority of the studies were conducted on adults with neurological disorders (e.g., multiple sclerosis, Parkinson’s disease, Huntington’s disease, stroke, and dementia) (32.4%) and adults with chronic conditions (e.g., chronic obstructive pulmonary disease, renal disease, osteoarthritis, metabolic syndrome, and fibromyalgia) (21.6%). Approximately 10% of these published articles focused on adults with cancer. Fifty-six percent (n = 21) of the published articles had at least 30 participants, with an average of 54 participants per study (Table 1).

Almost all reports (97%) were interventional study designs aimed at examining the effect of exercise/PA on BDNF and symptoms. Aerobic exercise (44%) and multimodal exercise (26.5%), combined aerobic activity, strength training, and stretching were the most-used exercise/PA modalities across studies. Approximately 35% of the studies (n = 13) progressively increased exercise intensity and duration to achieve moderate intensity. A small proportion of studies (n = 3) focused on personalized home-based exercise programs [48,49,50], and one study investigated the effects of increasing daily PA in free-living conditions [51]. 

Table 2 shows that the studies varied in sample sizes, ranging from 8 to 451 participants, and predominantly focused on adult populations, with some targeting older adults and specific subgroups. The gender distribution varied, with some studies including only female participants, such as women with fibromyalgia, while others had a more balanced or male-dominated composition. The mean age of participants generally ranged from the early 30s to mid-70s. Many studies included comparison groups, such as healthy controls, while others focused only on the intervention group.

**Table 2 biomedicines-13-00332-t002:** Study sample characteristics.

Author, Year of Publication	Country	Study Design	Participants	n	Gender (Male/Female)	Age (Range) Mean ± SD
1. Amato et al., 2021 [52]	Italy	Interventional design	Adults with MS	8	NA	34.88 ± 4.45
2. Azevedo et al., 2022 [53]	Brazil	Interventional design	Adults with PD	30	24/6	63.82 ± 9.63
3. Bansi et al., 2013 [54]	Switzerland	Interventional design	Adults with MS	52	18/34	51.08 ± 2.48
4. Bartlett et al., 2020 [55]	Australia	Interventional design	Adults with HD	29	10/19	44.55 ± 11.77
5. Devasahayam et al., 2020 [56]	Canada	Interventional design	Adults with MS	21	7/14	53.2 ± 15.6
6. Belchior et al., 2017 [57]	Brazil	Interventional design	Adults with PD	22	11/7	72.13 ± 12.10
7. Harro et al., 2022 [58]	USA	Interventional design	Adults with PD	12	8/4	67.17 ± 9.19
8. Landers et al., 2019 [59]	USA	Interventional design	Adults with PD	24	19/8	64.03 ± 8.74
9. Ozkul et al., 2018 [60]	Turkey	Interventional design	Adults with MS	54	12/42	MS = 34.25 ± 3.64 HC = 33 ± 4.13
10. Liu et al., 2020 [61]	Taiwan	Interventional design	Older adults with dementia	61	50/11	85.71 ± 6.81
11. Zhang et al., 2023 [62]	China	Interventional design	Older adults with MCI	42	4/38	(60–80)
12. De Araujo et al., 2019 [63]	Brazil	Interventional design	Adults with COPD	16	9/7	68.5 ± 6.7
13. Deus et al., 2021 [64]	Brazil	Interventional design	Adults with Renal disease undergoing hemodialysis treatments	157	86/71	66.81 ± 3.55
14. Gomes et al., 2014 [65]	Brazil	Interventional design	Women with knee OA	16	0/16	67 ± 4.41
15. Jablochkova et al., 2019 [66]	Sweden	Interventional design	Adults with FM	100	0/100	FM = 50.8 ± 9.6 HC = 47.6 ± 12.8
16. Lee et al., 2014 [67]	Taiwan	Interventional design	Adults with metabolic syndrome	36	36/0	44 ± 9.91
17. Ribeiro et al., 2021 [68]	Brazil	Interventional design	Women with FM	32	0/32	54 (50–58); 56 (53–59)
18. Žlibinaitė et al., 2020 [69]	Lithuania	Interventional design	Overweight and obese women (BMI > 25 kg/m^2^)	26	0/26	44.9 ± 6.2
19. Maguire et al., 2023 [70]	Switzerland	Interventional design	Adults with chronic ischemic or hemorrhagic stroke	17	13/4	55.12 ± 7.41
20. Cartmel et al., 2021 [48]	USA	Interventional design	Women with stage I–IV ovarian cancer	144	0/144	57.3 ± 8.6
21. Hartman et al., 2019 [49]	USA	Interventional design	Adults with breast cancer	87	0/87	57 ± 10.4
22. Miklja et al., 2022 [71]	USA	Correlational design	Adults with glioma	38	23/15	50
23. Zimmer et al., 2018 [50]	Germany	Interventional design	Adults with breast cancer	60	0/60	54.30 ± 8.51
24. Gmiat et al., 2018 [72]	Poland	Interventional design	Healthy older adults	35	0/35	69 ± 5.12
25. Pereira et al., 2013 [73]	Brazil	Interventional design	Community-dwelling older women	451	0/451	70.69 ± 4.66
26. Ruiz et al., 2015 [74]	Spain	Interventional design	Older adults in nursing homes	40	8/32	92.2 ± 2.27
27. Yeh et al., 2015 [75]	Taiwan	Interventional design	Community-dwelling women	67	0/67	52.7 ± 10.9
28. Takahashi et al., 2019 [51]	Japan	Interventional design	Postmenopausal women	38	0/38	70.2 ± 3.9
29. Vedovelli et al., 2017 [76]	Brazil	Interventional design	Older women	29	0/29	81.24 ± 8.0
30. Cahn et al., 2017 [77]	USA	Interventional design	Healthy adults	38	19/19	34.28 ± 8.84
31. Cullen et al., 2020 [78]	UK	Interventional design	Healthy adults	10	10/0	27 ± 6.0
32. Piacentini et al., 2016 [79]	UK	Interventional design	Healthy adults: endurance athletes (well-trained cyclists)	8	NA	27 ± 8.0
33. Suzuki et al., 2014 [80]	Japan	Interventional design	Healthy adults	52	52/0	26.6 ± 3.1
34. Verbickas et al., 2017 [81]	Lithuania	Interventional design	Healthy and physically active men	20	20/0	26.2 ± 8.21
35. Verbickas et al., 2018 [82]	Lithuania	Interventional design	Healthy adults	10	10/0	21.3 ± 2.3

Note: MS, multiple sclerosis; PD, Parkinson’s disease; HD, Huntington’s disease; MCI, mild cognitive impairment; COPD, chronic obstructive pulmonary disease; OA, osteoarthritis; FM, fibromyalgia; HC, healthy volunteers.

Details of the study intervention characteristics, measures, and outcomes are provided in the Appendix A. Program durations and frequencies varied across all studies. The program duration ranged from one (60 min) session to 72 exercise sessions (60 min/session). Ninety-four percent (n = 33) of the interventions employed were supervised by a trained professional and/or research staff. Many studies had control groups, including usual care or standard interventions. For example, some studies compared exercise groups with other types of activity, such as relaxation therapy, low-intensity training, or no intervention at all. Seven papers reported the immediate effect of exercise on symptoms and BDNF levels [62,63,65,76,80,81,82], while three studies investigated the long-term post-intervention effects at 3 to 9 months [52,58,59]. These long-term studies compared the post-intervention outcomes with baseline measurements to assess the sustainability of exercise benefits on BDNF and symptom improvement. The results were mixed, with some studies showing changes in BDNF levels, while others found no substantial differences between the intervention and control groups, particularly in long-term follow-ups. Symptoms (e.g., pain, fatigue, depression, sleep disturbance, anxiety) were studied as outcomes in exercise/PA programs. Depression (63%) and fatigue (46%) were the most often studied symptom outcomes of exercise/PA. More than 60% of the studies observed significant increases in BDNF levels following exercise/PA interventions, particularly after aerobic and multimodal exercise programs. Significant changes in symptoms were also reported in numerous studies, with improvements seen in fatigue, depression, pain, and quality of life. For instance, studies by Azevedo et al. (2022) and Gomes et al. (2014) [53,65] noted significant reductions in fatigue and pain, respectively. However, the relationship between changes in BDNF and symptom improvements was unclear. While some studies, such as those by De Araujo et al. (2019) and Deus et al. (2021) [63,64], reported significant correlations between BDNF levels and symptom changes like depression, others found no significant associations between BDNF changes and symptom improvements. Thus, while BDNF often increased after exercise, the connection between these changes and symptom reduction varied across studies, suggesting that other factors may influence symptom improvements beyond just BDNF levels.

All selected articles measured circulating BDNF in plasma (n = 13; 37.1%) or serum (n = 22; 62.9%). Only two studies explored specific variations in BDNF in the DNA sequence in addition to circulating BDNF levels [59,71]. While some studies did not report the timing of blood collection, 19 studies specified that the participants’ blood was collected in the morning or 8 to 12 hours after a fasting period (51%). Four studies asked participants to refrain from caffeine and alcohol consumption, sleep deprivation, and strenuous physical activity to prevent the impact of circadian rhythm on BDNF levels. Most studies explored the effects of exercise/PA programs on circulating BNDF levels (n = 31, 83.7%). Some studies investigated the relationship between the changes in BDNF levels and symptoms (n = 17; 45.9%).

Only three studies examined the mediating effect of BDNF on the effect of exercise/PA on symptoms (depression and cognitive dysfunction) [49,73] and grouping criteria (high vs. low BDNF groups) [67]. While 71.4% of studies (n = 25) noted increased circulating BDNF levels after exercise/PA, with 16 papers reporting significant results, six studies reported either a significant reduction [63,80] or no significant change [49,57,70,83].

The informetric analysis (Table 3) organizes the study findings of the 35 articles into four categories: symptom-related biomarkers, exercise/PA response, mediator, and inactive. Nearly 70% of the reviewed articles assessed BDNF as an outcome of an exercise/PA intervention, suggesting its potential role in mediating physiological changes associated with increased exercise/PA. Some articles (23%) sought to understand the role of BDNF as the symptoms’ underlying mechanism. Some articles (21%) found no significant relationship between BDNF and symptoms or physical activity. Only two articles (6%) explored the mediator role of BDNF on the effect of exercise on symptoms.

**Table 3 biomedicines-13-00332-t003:** BDNF roles.

Categories	Descriptions	% *	Reference
Symptom-related biomarkers	Indicating biophysical conditions (e.g., stress, depression, fatigue, and pain)	23	[48,56,62,64,65,78,79,80]
Exercise/physical activity response	Resulting from physical activities (e.g., aerobic, strengthening, intensified, or combination of exercises)	69	[48,50,51,52,53,54,59,60,62,63,64,65,67,68,71,72,73,75,76,77,78,80,81,82]
Mediator	Mediating the effect of exercise/physical activity on symptoms	6	[49,73]
Inactive	Showing no response to physical exercises (e.g., no changes in BDNF level) Displaying no relations to physical symptoms (e.g., no correlations with symptoms)	20	[55,58,61,66,69,70,74]

* One study may report more than one category. Percentages were calculated based on the total number of reports included.

We explored the possible mechanisms through which exercise/PA modulates BDNF expression and its subsequent effects on various symptoms, including fatigue, pain, depression, and sleep disturbances. Exercise-/PA-induced increases in BDNF levels were consistently associated with improvements in these symptoms, highlighting the complex relationship between physical activity, neuroplasticity, and symptom alleviation. Specifically, exercise/PA enhanced neuronal activity, synaptic plasticity, and neurogenesis, supporting physical and cognitive function, emotional regulation, and pain modulation (Figure 2).

## 4. Discussion

Although previous reviews have established BDNF’s role in neuroplasticity and brain health [36,38,39], a significant gap remains in understanding how exercise-/PA-induced BDNF changes help alleviate symptoms such as fatigue, pain, depression, and sleep disturbances in individuals without mental health conditions. Our scoping review findings support BDNF’s longstanding role as a marker of exercise/PA response since its discovery in 1982 [84,85] and suggest a potential connection between BDNF changes and symptom improvements, such as reductions in fatigue, pain, and depression. These findings reinforce BDNF’s role in neuroplasticity and neuroprotection, suggesting that exercise-related increases in BDNF may promote emotional well-being and symptom relief.

### 4.1. BDNF as a Marker for Fatigue, Pain, Depression, and Sleep

Some studies have reported positive correlations among exercise, BDNF, and symptom relief, particularly for depression. However, these findings are inconsistent, with variability across populations and exercise modalities, complicating clinical applications. For instance, Amato et al. (2021) [52] observed a significant increase in BDNF levels after lactate threshold training, which correlated with reduced fatigue. However, this effect did not persist at a 9-month follow-up, suggesting that while exercise can elevate BDNF levels, the long-term impact of BDNF reduction on fatigue relief may be limited. Similarly, Bansi et al. (2013) [54] found increased BDNF levels after aquatic cycling but no significant reduction in fatigue, indicating that BDNF elevation does not always correlate with symptom improvement. Devasahayam et al. (2020) [56] found increased BDNF levels after a graded exercise test in multiple sclerosis patients. Still, the change was insufficient to reduce fatigue, although higher BDNF levels were associated with better walking speeds. These findings highlight the complex relationship between BDNF and fatigue, where changes in BDNF do not always align with improvements in fatigue.

Several studies have examined the relationship between exercise-induced BDNF changes and pain relief, with mixed findings. For instance, Gomes et al. (2014) [65] found that a 12-week aerobic exercise program significantly increased BDNF levels, which were associated with pain reduction. Similarly, Ribeiro et al. (2021) [68] observed that a 6-week whole-body vibration training program increased BDNF levels, correlating with reduced pain and improved sleep quality in individuals with fibromyalgia. However, other studies, such as Jablochkova et al. (2019) [66], reported no significant association between changes in BDNF levels and pain reduction despite improved pain and fatigue following resistance training. Additionally, Verbickas et al. (2018) [82] found that BDNF levels decreased post-exercise, with no observed relationship between BDNF changes and peripheral or central fatigue. These findings suggest that while BDNF may be involved in pain modulation, its role as a consistent biomarker for pain relief remains uncertain.

Regarding depression, Azevedo et al. (2022) [53] found that a single session of aerobic exercise increased BDNF, but the effect on depression was not significant. Similarly, Landers et al. (2019) [59] reported increases in BDNF post-intervention, but depression reduction was not consistently associated with changes in BDNF, suggesting that other factors may contribute to symptom relief. These inconsistencies suggest that BDNF may not be a reliable marker for depression relief across all individuals or exercise. While BDNF may play a role in exercise-induced neuroplasticity, its use as a consistent biomarker for symptom relief remains unclear. Variability in BDNF responses across different exercise regimens makes applying BDNF as a reliable symptom indicator challenging. Further research is needed to clarify when exercise-induced BDNF changes are most likely to result in symptom relief, particularly for fatigue, pain, and depression.

Exercise has been shown to increase BDNF levels; however, its effects on sleep quality remain inconsistent. For example, Bartlett et al. (2020) [55] conducted a 9-month intervention that included aerobic, resistance, and endurance training. Although there was a slight increase in BDNF levels, no significant improvements in sleep quality were observed. Similarly, Devasahayam et al. (2020) [56] reported an increase in BDNF levels after a graded exercise test, but no improvement in sleep-related measures was noted. In contrast, Ribeiro et al. (2021) [68] observed that a 6-week whole-body vibration training program increased BDNF levels and improved sleep quality, pain, and depression in individuals with fibromyalgia. These findings suggest that, in certain populations, BDNF may play a role in improving sleep quality following specific exercise interventions. The variability in findings across studies reflects the complexity of the relationship between exercise, BDNF, and sleep. Factors such as exercise modality, intensity, and baseline sleep conditions may influence how BDNF affects sleep disturbances.

### 4.2. BDNF as a Mechanistic Biomarker

BDNF has attracted attention as a potential mechanistic biomarker for symptom relief, especially in fatigue, pain, depression, and sleep disturbance. Evidence suggests that exercise-induced BDNF contributes to symptom improvements through several interconnected mechanisms. BDNF enhances neuroplasticity by promoting synaptogenesis, dendritic growth, and long-term potentiation (LTP), which are essential for learning, memory, and emotional regulation [86]. It also stimulates neurogenesis, particularly in the hippocampus, a brain region critical for mood regulation and cognitive function. Additionally, BDNF interacts with neurotransmitter systems such as serotonin, dopamine, and glutamate, enhancing neurotransmitter availability and receptor sensitivity, thereby improving mood and reducing symptoms of depression and anxiety [87]. Exercise-induced increases in BDNF also help regulate inflammatory pathways by reducing pro-inflammatory cytokines and fostering an anti-inflammatory environment, which is crucial in alleviating symptoms of fatigue and depression. Moreover, BDNF supports mitochondrial function and energy metabolism, improving cellular energy production and reducing fatigue [88]. It is vital in regulating the hypothalamic–pituitary–adrenal (HPA) axis, lowering cortisol levels, and enhancing stress resilience. Finally, exercise enhances cerebral blood flow, which, together with BDNF, supports vascularization and oxygenation, promoting overall brain health and cognitive function [89].

While exercise-induced increases in BDNF are associated with improvements in symptoms like fatigue and depression, its role as a mechanistic biomarker linking BDNF changes to symptom improvement remains inconsistent. For example, Amato et al. (2021) [52] found that BDNF levels increased immediately after lactate threshold training, correlating with reduced fatigue. Similarly, De Araujo et al. (2019) [63] reported increased BDNF levels after treadmill endurance training. Still, changes in BDNF did not correspond to significant improvements in depression or dyspnea, suggesting that BDNF does not always mediate symptom relief. The variability in BDNF responses across different exercise modalities, intensities, and populations further complicates its role as a mechanistic biomarker. Moreover, BDNF levels can fluctuate rapidly. Studies such as those conducted by Devasahayam et al. (2020) and Amato et al. (2021) [52,56] showed that although BDNF increases immediately after exercise, these changes may return to baseline levels quickly, raising concerns about BDNF’s long-term relevance as a marker for sustained symptom relief.

Variability in BDNF responses has also been observed in pharmacological studies investigating its potential as a mechanistic biomarker for symptom improvement, particularly in depression. For example, studies on glutamatergic agents such as ketamine and dextromethorphan found that ketamine significantly increased pro-BDNF expression in the hippocampus within 40 min via an AMPA receptor-dependent mechanism—a response not observed with dextromethorphan [90]. While ketamine’s BDNF-mediated effects correlate with its rapid antidepressant outcomes, the absence of significant BDNF changes following dextromethorphan treatment suggests the involvement of alternative pathways in its therapeutic effects. One such pathway is the sigma-1 receptor, which plays a crucial role in neuroprotection by regulating calcium homeostasis, reducing oxidative stress, and modulating endoplasmic reticulum (ER) stress [6]. Sigma-1 receptor agonists have been shown to stabilize mitochondrial function and enhance neurotrophic support [91], suggesting a potential interaction with BDNF pathways in improving symptoms. However, further research is needed to clarify whether BDNF changes directly drive symptom relief or if they are secondary to broader neuroprotective processes.

### 4.3. BDNF as a Mediator in Symptom Improvement

An emerging hypothesis is that BDNF may mediate the relationship between exercise/PA and symptom improvement. While this idea remains underexplored, some studies suggest that BDNF could play a role in mediating improvements in conditions like depression, fatigue, and cognitive dysfunction. For instance, Deus et al. (2021) [64] found that resistance training increased BDNF levels, which were associated with improved depression scores and emotional well-being in individuals undergoing hemodialysis. Similarly, Ribeiro et al. (2021) [68] observed reductions in pain, sleep disturbances, and depression following whole-body vibration training, with concomitant increases in BDNF levels. However, the mediating role of BDNF is not universally supported. Gomes et al. (2014) [65] found that although BDNF levels increased after 12 weeks of chronic exercise, pain decreased only in the intervention group and did not correlate with BDNF changes. Similarly, Bansi et al. (2013) [54] found an increase in BDNF after aquatic cycling but no corresponding reduction in fatigue, suggesting that BDNF may not always mediate symptom improvement.

The variability in findings highlights the complexity of BDNF’s role as a mediator. Factors such as exercise type, intensity, duration, and individual characteristics (e.g., baseline health status and age) may influence BDNF’s mediating effect. Harro et al. (2022) [58] found that while BDNF levels increased after Nordic walking, there was no significant change in fatigue, suggesting that exercise modality may impact BDNF’s mediating role. Additionally, the temporal dynamics of BDNF levels complicate its role as a long-term mediator, as BDNF increases are often transient. Individual variability also plays a crucial role in BDNF’s potential as a mediator. Miklja et al. (2022) [71] found that exercise tolerance was inversely associated with BDNF levels in glioma patients, indicating that BDNF’s role may vary based on individual characteristics and health conditions.

### 4.4. Limitations and Strengths

One limitation of this scoping review is the decision to restrict the database search to only PubMed and CINAHL Plus. While these databases provided substantial coverage of the relevant literature in biomedical and healthcare research, using a limited number of databases may have resulted in the omission of relevant studies available in other databases. Expanding the search strategy to include additional databases could have captured a wider range of studies, increasing the comprehensiveness of the review.

Another significant limitation of our review is the variability in the study designs included, which challenges direct comparisons across studies. It is a strength that a scoping review allows for a broad assessment of the existing literature. However, it does not involve a thorough evaluation of study methodologies, risk of bias, and the strength of the evidence, which would allow for more precise conclusions. In our review, the studies varied widely regarding exercise/PA protocols, including differences in the type of exercise, frequency, duration, and intensity. Furthermore, the populations studied were diverse, including varying age groups, health statuses, and underlying conditions, which limits the ability to generalize the findings across different settings or groups. Another complication arises from the inconsistent methods used to measure BDNF levels across the studies. Variations in blood sampling timing, analysis methods, and BDNF assay types introduced significant methodological heterogeneity. These discrepancies make it difficult to interpret and synthesize the results in a meaningful way, ultimately limiting the conclusions that can be drawn about the relationship between exercise/PA and BDNF levels.

Furthermore, interpreting the relationship between BDNF levels and their biological activity is complex and limited. While many studies report significant increases in BDNF following exercise interventions, the mere elevation in BDNF does not always correlate with its biological activity or the activation of specific neurophysiological pathways. For instance, treatments that influence BDNF receptor ligands may alter BDNF levels without necessarily enhancing its ability to activate downstream pathways or trigger physiological responses. Additionally, genetic variations, such as the *BDNF Val66Met* polymorphism, can affect the functional activity of BDNF, potentially influencing symptom outcomes even in the presence of unchanged BDNF levels. Also, alternative pathways beyond direct BDNF signaling may play significant roles in mediating these effects, such as AMPA receptor potentiation and the activation of intracellular signaling cascades like mTOR and Akt, which can mimic the behavioral and biochemical outcomes typically linked to BDNF enhancement [92]. These factors may contribute to the variability in BDNF response to exercise and underscore the need for more research to investigate the complex interactions between BDNF levels, its activity, and genetic factors in determining the therapeutic effects of exercise.

Our review included only studies conducted on human subjects, which limits the generalizability of the findings to animal models or studies involving non-human organisms. The criteria for inclusion also restricted the studies to those involving healthy volunteers or individuals with physical conditions such as non-communicable chronic conditions (e.g., cancer, neurological disorders, renal disease), excluding studies on mental health conditions (e.g., schizophrenia, bipolar disorder, depression, anxiety, eating disorders) and children under the age of 18. Studies published in languages other than English were excluded, limiting the review’s comprehensiveness. Additionally, the review lacks experimental article data, which would have provided more substantial evidence for causal relationships. This absence limits the ability to definitively conclude the mechanisms through which exercise impacts BDNF levels and associated symptoms. These factors should be considered when interpreting the findings of this review.

Despite these limitations, this scoping review has several strengths. It provides a comprehensive overview of the current research landscape on BDNF and exercise/PA, offering valuable insights into gaps in the literature and identifying areas for future exploration. Including diverse populations and exercise/PA modalities highlight trends and inconsistencies that may not be visible in more narrowly focused reviews. Additionally, this review underscores the importance of BDNF as a biomarker for exercise/PA response and suggests its potential role in symptom management.

### 4.5. Implications for Future Research

These findings emphasize the need for further research to address the variability in exercise/PA interventions and BDNF measurement techniques. Standardizing exercise interventions and BDNF assessment methods across studies is crucial for establishing its clinical relevance as a biomarker for symptom management. Future research should prioritize using consistent protocols and validated BDNF measurement techniques to ensure reliable and comparable data. Beyond standardization, it is essential to investigate the specific mechanisms through which BDNF influences symptom relief to better translate findings into clinical practice and identify the most effective exercise regimens for individuals with various chronic conditions. While exercise-induced BDNF changes provide promising insights into symptom management, their transient nature and variability raise concerns about their long-term utility as a mechanistic biomarker, further emphasizing the need for research to elucidate its mechanistic role and develop standardized methodologies for reliable measurement in clinical and research settings. Addressing these gaps will help clarify BDNF’s potential as a therapeutic target, ultimately leading to more personalized and effective interventions for symptom management.

## 5. Conclusions

In conclusion, this scoping review highlights the limited but notable evidence supporting BDNF as a potential marker for the effectiveness of exercise/PA in managing symptoms such as fatigue, pain, depression, and sleep disturbances. While many studies reported increases in BDNF levels, less than half showed significant increases in BDNF following exercise or physical activity programs, and only a few demonstrated a significant relationship between changes in BDNF and symptoms. Despite these findings, consistent associations between BDNF changes and symptom improvement were not observed across studies. Furthermore, no significant differences in BDNF response were found across various types of exercise or PA. These findings suggest the need for further investigations and randomized controlled trials to elucidate the impact of exercise/PA on BDNF levels and the direct effects of BDNF on symptom management.

## Figures and Tables

**Figure 1 biomedicines-13-00332-f001:**
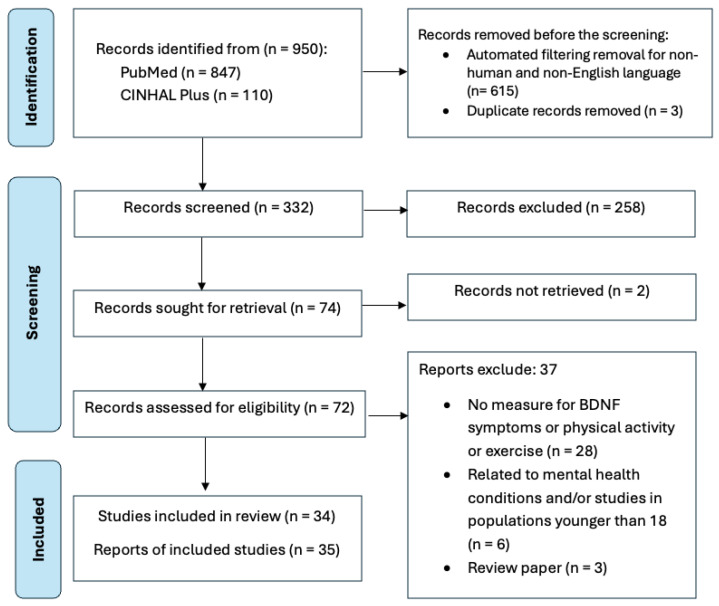
PRISMA flow diagram of study selection. Adapted from the PRISMA-ScR diagram [45].

**Figure 2 biomedicines-13-00332-f002:**
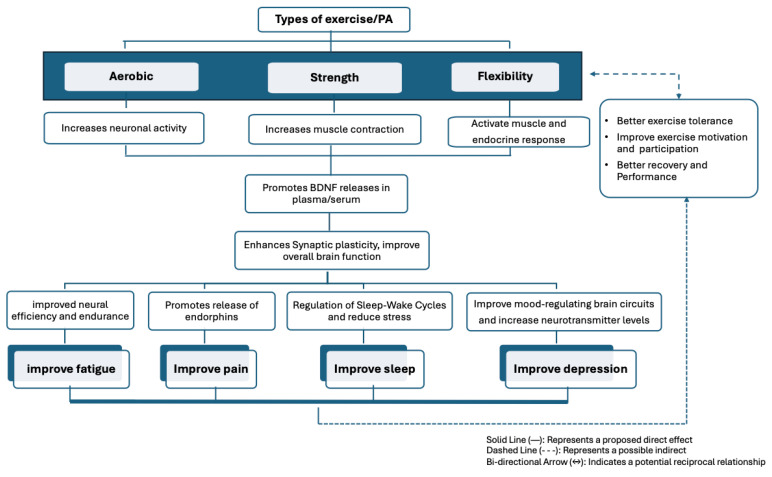
Exercise-/physical activity-induced BDNF modulation and its impact on fatigue, pain, sleep, and depression.

**Table 1 biomedicines-13-00332-t001:** Study characteristics (n = 35).

Study Characteristics	n	%
Publication year (Mean = 2019, SD = 2.9)		
2013–2015	7	20
2016–2018	9	25.7
2019–2021	14	40
2022–2024	5	14.3
Study region *		
The Americas	15	42.9
Europe	13	37.1
Western Pacific	7	21.6
Study populations		
Adults with neurological disorders	10	28.6
Adults with chronic conditions	9	25.7
Adults with cancer	4	11.4
Older adults (aged 60 and older)	6	17.1
Healthy young adults	6	17.1
Sample size ** (n = 1882; Mean = 53.8; SD = 77.3)		
8–29	15	42.9
30–59	12	34.3
60–99	4	11.4
100 and above	4	11.4
Study design		
Interventional study	34	97.1
Correlational study	1	2.9
Exercise/physical activity types reported in interventional studies (n = 34)		
Aerobic exercise and endurance training	15	44.1
Strength training/resistance	4	11.8
Flexibility/stretching (e.g., yoga, etc.)	2	5.88
Multimodal exercise	9	26.5
Others (i.e., whole body vibration, drop jumps)	2	5.88
Physical activity	2	5.88
Symptoms reported ^†^		
Pain	6	17.1
Fatigue	16	45.7
Depression	22	62.9
Anxiety	6	17.1
Moods (overall moods)	3	8.6
Sleep disturbance	4	11.4
Cognitive function	7	20.0
Dyspnea	1	2.9
BDNF ^†^		
Circulating BDNF		
- Plasma level	13	37.1
- Serum level	22	62.9

* The region was based on the six World Health Organization regions. ** Sample size: n represents the total number of participants across all studies, where M indicates the mean sample size and SD represents the standard deviation. ^†^ One study may report more than one category. Percentages were calculated based on the total number of reports included.

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
