# Peer review of "Brain-Derived Neurotrophic Factor (BDNF) as a Marker of Physical Exercise or Activity Effectiveness in Fatigue, Pain, Depression, and Sleep Disturbances: A Scoping Review"

_biomedicines, 2025, doi:10.3390/biomedicines13020332_

Round 1
Reviewer 1 Report
Comments and Suggestions for Authors
This scoping review by Nada Lukkahatai and collaborators aims to explore the potential role of BDNF as a biomarker in exercise-based interventions for symptom management. Overall, the manuscript is well-developed; however, the following recommendations must be addressed for it to be considered for publication:
Major Corrections:
1. The focus on "Symptom Management" is too broad. It is recommended that the title and abstract be adjusted to specifically reflect the disorders addressed, such as fatigue, pain, depression, and sleep.
2. I suggest the authors include a figure integrating the mechanisms by which exercise modulates BDNF expression and its influence on symptomatology. This would enhance the manuscript's analysis.
3. Since the methods include the keywords “exercise OR physical activity,” it is necessary to adjust the title and sections of the text where these terms are used interchangeably to maintain consistency.
4. The manuscript should clarify why only these four symptoms (fatigue, pain, depression, sleep) were selected and why the inclusion was limited to articles published between 2010 and 2024.
5. The distinction between "depression" and "clinical depression," as well as between "anxiety" and "anxiety disorders," is unclear. Please specify these definitions to avoid confusion.
6. The table is difficult to read and requires further synthesis. The last column and ELISA kit characteristics appear irrelevant. It is important to include participants' gender and, if applicable, information about the controls and their characteristics.
7. The conclusion should be reinforced by emphasizing the importance of BDNF as a biomarker and its relationship with exercise's benefits on symptomatology.
Minor Corrections:
1. Add a brief paragraph about the brain areas where BDNF is predominantly expressed to provide context for its relevance.
2. Synthesize Lines L62 to L90. This section is redundant. Synthesizing the information is recommended for greater clarity.
3. Term in L128 "psychological disorders" is inappropriate in this context and should be corrected.
4. The figure 1 lacks a descriptive legend. Additionally, it mentions excluding 33 reports; verify whether this number is correct.
5. In Table 2 studies 8, 21, and 34 do not evaluate the symptoms outlined in the inclusion criteria (fatigue, pain, depression, sleep). Their relevance should be reviewed.
6. In the L233, verify this line's wording to ensure its accuracy.
Author Response
Response letter
Journal: Biomedicines (ISSN 2227-9059)
Manuscript ID: biomedicines-3320875
Type: Review
Title: Brain-Derived Neurotrophic Factor (BDNF) as a marker of Physical Exercise Effectiveness in Symptom Management:
A Scoping Review
|
Comment |
Response |
Page/Line |
|
Reviewer 1 |
||
|
Comment 1: This scoping review by Nada Lukkahatai and collaborators aims to explore the potential role of BDNF as a biomarker in exercise-based interventions for symptom management. Overall, the manuscript is well-developed; however, the following recommendations must be addressed for it to be considered for publication: |
Response 1: Thank you for your thoughtful comments and recognition of our work. We will carefully address your recommendations to enhance the manuscript and meet the publication standards. |
- |
|
Major Corrections: Comment 2: The focus on "Symptom Management" is too broad. It is recommended that the title and abstract be adjusted to specifically reflect the disorders addressed, such as fatigue, pain, depression, and sleep. |
Response 2: Thank you for your valuable feedback. To address this, we have revised the title, abstract and discussion to specifically highlight the disorders discussed in the paper, namely fatigue, pain, depression, and sleep disturbances. This more focused approach ensures clarity in the paper’s objectives and aligns the discussion with the primary symptoms explored in relation to BDNF and exercise. The revised title now reads: "Brain-Derived Neurotrophic Factor (BDNF) as a Marker of Physical Exercise or Activity Effectiveness in Managing Fatigue, Pain, Depression, and Sleep Disturbances: A Scoping Review." Additionally, the abstract and discussion section have been updated to reflect this shift in focus, emphasizing the relationship between exercise, BDNF, and the specific symptoms mentioned. We believe these changes will improve the precision of the paper's scope and enhance its relevance to readers. |
Page 1/ Line 2-3, 16-29 |
|
Comment 3: I suggest the authors include a figure integrating the mechanisms by which exercise modulates BDNF expression and its influence on symptomatology. This would enhance the manuscript's analysis. |
Response 3: We have created a diagram (Figure 2: Exercise/PA-induced BDNF Modulation and its impact on fatigue, pain, sleep, and depression) that illustrates the key processes by which exercise increases BDNF levels and how these changes contribute to symptom relief in fatigue, pain, depression, and sleep disturbances in result section |
Page 21/ Line311-313 |
|
Comment 4: Since the methods include the keywords “exercise OR physical activity,” it is necessary to adjust the title and sections of the text where these terms are used interchangeably to maintain consistency. |
Response 4: We have adjusted the title and relevant sections of the manuscript to ensure that both terms are used consistently throughout. In particular, we have updated the title and specific sections of the text where these terms are used and add a statement in the method section (Page 3, Line 125-127) to reflect their interchangeable usage. |
Page 1/ Line 2-3 |
|
Comment 5: The manuscript should clarify why only these four symptoms (fatigue, pain, depression, sleep) were selected and why the inclusion was limited to articles published between 2010 and 2024. |
Response 5: Thank you for your insightful feedback. We selected fatigue, pain, depression, and sleep disturbances as the primary symptoms for this review due to their high prevalence in individuals with chronic conditions and their significant impact on quality of life. We include this justification in the introduction (Page 3, Line 101-108) For the year limit, upon reviewing the manuscript, we realize that we did not restrict the publication year of the included articles. The information on publication years was corrected in the method section (Page 3, Line 133-134) |
Page 3/Line 101-108 Page 3/Line 133-134 |
|
Comment 6: The distinction between "depression" and "clinical depression," as well as between "anxiety" and "anxiety disorders," is unclear. Please specify these definitions to avoid confusion. |
Response 6: Thank you for this comment. We have added further clarification in the Introduction section to provide context and ensure the terms are clearly defined (Page 3, Line 106-116). Additionally, we have included precise definitions in the Methods section (Page 4, Line 157-158, and 166) to avoid any confusion and to ensure consistency throughout the manuscript. |
Page 3, Line 106-116 Page 4, Line 158-459 and 165 |
|
Comment 7: The table is difficult to read and requires further synthesis. The last column and ELISA kit characteristics appear irrelevant. It is important to include participants' gender and, if applicable, information about the controls and their characteristics. |
Response 7: We have revised the table to enhance clarity and improve readability. The table has been separated into two distinct tables: table 2 (Page 7/ Line 243) for study sample characteristics to include gender and characteristics of control groups for each study and table 3 (Page 10/ Line 259) for intervention, outcomes and key findings with additional result information. Additionally, the columns regarding the ELISA kit characteristics and irrelevant details have been removed. |
Page 7/ Line 243 Page 10/ Line 259 |
|
Comment 8: The conclusion should be reinforced by emphasizing the importance of BDNF as a biomarker and its relationship with exercise's benefits on symptomatology. |
Response 8: Thank you for your suggestion. We have updated the conclusion to emphasize the importance of BDNF as a biomarker and its relationship with the benefits of exercise on symptomatology. |
Page 25/ Line 482-492 |
|
Minor Corrections: Comment 9: Add a brief paragraph about the brain areas where BDNF is predominantly expressed to provide context for its relevance. |
Response 9: Thank you for your suggestion. We have added a brief paragraph to the introduction regarding the brain areas where BDNF is predominantly expressed, including the hippocampus, cortex, basal forebrain, and amygdale in the introduction (Page 1-2/ Line 41-49). |
Page 1-2/ Line 41-49 |
|
Comment 10: Synthesize Lines L62 to L90. This section is redundant. Synthesizing the information is recommended for greater clarity. |
Response 10: Thank you for your suggestion. We have synthesized (Page 2/ Lines 62-90, old version) to remove redundancy and improve clarity. The revised section now presents the information more concisely while maintaining its key points. |
Page 2/ Line 87-99 |
|
Comment 11: Term in L128 "psychological disorders" is inappropriate in this context and should be corrected. |
Response 11: Thank you for pointing out the inappropriate use of the term "psychological disorders." We have updated it to "mental health conditions" throughout the manuscript in alignment with the terminology recommended by the WHO article. |
Page 4/ Line 157-158 |
|
Comment 12: The figure 1 lacks a descriptive legend. Additionally, it mentions excluding 33 reports; verify whether this number is correct. |
Response 12: Thank you for your feedback. We have updated the descriptive legend to: "Figure 1. PRISMA flow diagram of study selection and corrected the number of excluded reports in the figure to ensure accuracy. |
Page 5/ Line 207 |
|
Comment 13: In Table 2 studies 8, 21, and 34 do not evaluate the symptoms outlined in the inclusion criteria (fatigue, pain, depression, sleep). Their relevance should be reviewed. |
Response 13: Thank you for your valuable feedback. Upon review, we have confirmed that Study 34 included depression as an outcome, which aligns with our inclusion criteria. We removed study 8 (Hsu et al., 2021) and study 21 (Stein et al., 2023) and updated all tables, abstract and result sections to ensure the accuracy and consistency. |
- |
|
Comment 14: In the L233, verify this line's wording to ensure its accuracy. |
Response 14: Thank you for pointing out the potential assumption in the original wording. We have revised the sentence to: "Nearly 70% of reviewed articles assessed BDNF as an outcome of an exercise/PA intervention, which suggested its potential role in mediating physiological changes associated with increased exercise/PA program." This adjustment ensures a more accurate and neutral interpretation of the findings. |
Page 20/ Line 293-298 |
Reviewer 2 Report
Comments and Suggestions for Authors
The review is on a highly relevant topic of BDNF role as a marker in various physiological studies. I hope there will be more reviews like this one, showing that there is not enough evidence to confirm the existing consensus on some topic, despite most of the work here seems to be done using commercial software (Covidence). Although various details should be included in the manuscript to clarify what authors actually did.
1) Title should include which symptoms exactly can (or cannot?) be managed when measuring BDNF
2) Abstract and text in general should include some details on Covidence, at least that this is commercial software not some “scale” or “indicator” as I thought initially.
3) Abstract should include criteria for selection of papers (“Of 768 records, 37 records were 25 included”). So, please try to add some short info from the paragraphs 2.1 and 2.2.
4) The level of protein doesn’t always correlate with the protein biological activity. Did you check whether some treatments (maybe using BDNF receptor ligands) alter BDNF levels but do not affect its ability to activate some pathways or physiological responses (or vice versa)? Plus, some genetic variations (whose studies are actually included) can definitely affect the BDNF “activity” without altering its levels.
5) Paragraph 2.2 contains repeated selection criteria. You probably should not apply the same criteria for full-text analysis if you applied them during initial filtration. Also please clarify “studies in health volunteers or individuals with physical chronic conditions”. So healthy volunteers or not? And what is “physical chronic conditions”?
6) It’s unclear whether paragraph 2.3 is a framework within Covidence or work performed manually by authors. Probably latter, but please indicate this clearly. The same goes for description before Table 3 and Table 3 itself.
7) The main text should include some frequent reasons for rejection (young age? studies on animals? studies without measuring pain/mood/etc.? studies without exercises?) and how much papers were rejected during analysis of abstracts and how much – during analysis of full texts.
8) Table 2 should be transferred to Supplementary. Instead, some of the important findings (“Results” in this Table) should be presented as Figure. For example, as pie charts (showing percentages of studies where BDNF is increased/decreased and percentages of studies where BDNF correlates with physiological characteristics) or as forest plots. Because Table 3 is not “Results” but some processed results (plus, sum % is not 100% in this Table).
9) Lines 253-254 should cite some reference(s) or be removed.
10) Well, 67% of studies showing correlations of BDNF with physiology is a lot, so probably this information should be included both in Abstract and Conclusion.
And finally, since Covidence is a commercial software, there is always a potential conflict of interest. Please indicate, whether someone from your university is affiliated with the developers, and if not, how exactly did you or your organization get the access to the platform since “This research received no external funding”. Which seems impossible.
There are many grammar problems with spaces and commas, as well as starting the sentences with small letters. Please check your text before uploading.
Also, please indicate all authors in the References list, I suggest using Endnote with MDPI style.
Author Response
Response letter
Journal: Biomedicines (ISSN 2227-9059)
Manuscript ID: biomedicines-3320875
Type: Review
Title: Brain-Derived Neurotrophic Factor (BDNF) as a marker of Physical Exercise Effectiveness in Symptom Management:
A Scoping Review
|
Comment |
Response |
Page/Line |
|
Reviewer 2 |
||
|
The review is on a highly relevant topic of BDNF role as a marker in various physiological studies. I hope there will be more reviews like this one, showing that there is not enough evidence to confirm the existing consensus on some topic, despite most of the work here seems to be done using commercial software (Covidence). Although various details should be included in the manuscript to clarify what authors actually did. |
Thank you for your positive feedback and for recognizing the relevance of our review. We appreciate your observation about the need for clarity in detailing our methodology. We updated the methodology to include detail search to ensure that additional details regarding our processes are clearly outlined in the manuscript. |
Page 4/L187-193 |
|
1) Title should include which symptoms exactly can (or cannot?) be managed when measuring BDNF |
Thank you for your valuable feedback. We have revised the title, abstract and discussion to specifically highlight the disorders discussed in the paper, namely fatigue, pain, depression, and sleep disturbances. This more focused approach ensures clarity in the paper’s objectives and aligns the discussion with the primary symptoms explored in relation to BDNF and exercise. The revised title now reads: "Brain-Derived Neurotrophic Factor (BDNF) as a Marker of Exercise Effectiveness in Managing Fatigue, Pain, Depression, and Sleep Disturbances: A Scoping Review." Additionally, the abstract and discussion section have been updated to reflect this shift in focus, emphasizing the relationship between exercise, BDNF, and the specific symptoms mentioned. We believe these changes will improve the precision of the paper's scope and enhance its relevance to readers. |
Page 1/ Line 2-3 Page 21-23/ Line 324-401 |
|
2) Abstract and text in general should include some details on Covidence, at least that this is commercial software not some “scale” or “indicator” as I thought initially. |
Thank you for your comment. We have revised the text to replace "Covidence" with "Systematic review software" on the abstract (Page 1/ Line 22-23) and add an explanation of its role and usage in the Methods section (Page 3/ Line148-150). |
Page 1/ Line 22-23 Page 3/Line 148-150 |
|
3) Abstract should include criteria for selection of papers (“Of 768 records, 37 records were 25 included”). So, please try to add some short info from the paragraphs 2.1 and 2.2. |
Thank you for your comment. We have revised the abstract to include the selection criteria as shown below. |
Page 1/ Line 22 - 26 |
|
4) The level of protein doesn’t always correlate with the protein biological activity. Did you check whether some treatments (maybe using BDNF receptor ligands) alter BDNF levels but do not affect its ability to activate some pathways or physiological responses (or vice versa)? Plus, some genetic variations (whose studies are actually included) can definitely affect the BDNF “activity” without altering its levels. |
Thank you for the valuable comment. We fully acknowledge that the level of BDNF protein does not always correlate with its biological activity. As you suggested, some treatments and genetic variations may play role in BDNF activity without altering its protein levels. As information in all our reviewed papers for these factors were not available, we have included this important consideration in the limitations section of the manuscript (Page 24/ Line 435-446). |
Page 24 /Line 435-446 |
|
5) Paragraph 2.2 contains repeated selection criteria. You probably should not apply the same criteria for full-text analysis if you applied them during initial filtration. Also please clarify “studies in health volunteers or individuals with physical chronic conditions”. So healthy volunteers or not? And what is “physical chronic conditions”? |
Thank you for your insightful comments. We have applied broader criteria during the title and abstract screening and more specific criteria during the full-text analysis. Regarding "healthy volunteers or individuals with physical chronic conditions," we have corrected "health volunteer" to "healthy volunteer" for accuracy. Additionally, we included studies involving both healthy volunteers and individuals with physical chronic conditions while excluding those with mental health conditions. |
Page 4/ Line 154-156 |
|
6) It’s unclear whether paragraph 2.3 is a framework within Covidence or work performed manually by authors. Probably latter, but please indicate this clearly. The same goes for description before Table 3 and Table 3 itself. |
Thank you for highlighting this point. To clarify, we have revised the content to explicitly state: “The data extraction and charting process was manually extracted by NL and ILO and organized into a summary of evidence table format. The summary of the evidence table was then reviewed and iteratively refined by all authors to ensure comprehensive data extraction across studies.” (Page 4/ Line 183-186)
For table 4 BDNF role (previously table 3), we include more detail description of the classification process as stated “After generating results from Tables 1-3, the potential roles of BDNF (table 4) were identified as: 1) Symptoms-related Biomarkers (linking BDNF levels with symptoms), 2) Exercise/PA responses, 3) Mediator, and 4) Inactive (no correlation). Two reviewers (NL and ILO) independently classified the BDNF role for each study based on the study aims, methods, and results. The results were compared between the two reviewers, and any discrepancies were resolved through discussion. If a consensus could not be reached, a third reviewer (LNS) was consulted to ensure an accurate classification.” (Page 4/ Line 187-193) |
Page 4/Line 170 Page 4/Line 183-186
Page 4/Line 187-193 |
|
7) The main text should include some frequent reasons for rejection (young age? studies on animals? studies without measuring pain/mood/etc.? studies without exercises?) and how much papers were rejected during analysis of abstracts and how much – during analysis of full texts. |
We have revised the main text to include the frequent reasons for rejection in the 3.1 paragraph as shown below. |
Page 4/ Line 199-203 |
|
8) Table 2 should be transferred to Supplementary. Instead, some of the important findings (“Results” in this Table) should be presented as Figure. For example, as pie charts (showing percentages of studies where BDNF is increased/decreased and percentages of studies where BDNF correlates with physiological characteristics) or as forest plots. Because Table 3 is not “Results” but some processed results (plus, sum % is not 100% in this Table). |
Thank you for your feedback. To enhance clarity and improve readability, we have separated the original Table 2 into two tables: Table 2 (Page 7/ Line 243) now presents study sample characteristics, while Table 3 (Page 10/ Line 259) focuses on interventions, outcomes, and key findings. Additionally, we created a new diagram (Figure 2: Exercise/PA-induced BDNF Modulation and its Impact on Fatigue, Pain, Sleep, and Depression) (Page 21/ Line 312-313) to illustrate how exercise increases BDNF levels and contributes to symptom relief, aligning with the main purpose of our study. For Table 3, we added a footnote: "* One study may report more than one category. Percentages were calculated based on the total number of included reports," to clarify why the sum of percentages exceeds 100%. |
Page 7, Line 243 Page 10, Line 259 Page 21/ Line 312-313 |
|
9) Lines 253-254 should cite some reference(s) or be removed. |
Thank you for your comment. We have reviewed lines 253-254 and added appropriate references to support the statements. |
Page 21/ Line 315-318 |
|
10) Well, 67% of studies showing correlations of BDNF with physiology is a lot, so probably this information should be included both in Abstract and Conclusion. |
Thank you for your insightful suggestion. Upon the revision process, we were able to confirm that 74% (n=26) of the selected studies report increasing BDNF after exercise but only 40% (n=16) showed significant result. We include the statements “While exercise/PA is broadly supported for managing symptoms, and 74.3% (n=26) of studies reported increased BDNF levels, only 40% (n=14) showed significant increases following exercise/PA. Only 14% (n=5) of studies demonstrated a significant relationship between changes in BDNF and symptoms” in the abstract (Page 1/Line 26-29) and a statement “While many studies reported increases in BDNF levels, less than half showed significant increases in BDNF following exercise or physical activity programs, and only a few demonstrated a significant relationship between changes in BDNF and symptoms.” conclusion (Page 25/ Line 485-488) |
Page 1/Line 26-29 Page25/Line 485-488
|
|
And finally, since Covidence is a commercial software, there is always a potential conflict of interest. Please indicate, whether someone from your university is affiliated with the developers, and if not, how exactly did you or your organization get the access to the platform since “This research received no external funding”. Which seems impossible. |
Thank you for raising this concern. To clarify, neither the authors nor anyone from our university is affiliated with the developers of Covidence. Access to the Covidence platform was provided through an institutional subscription funded internally by our university, which does not constitute external funding. We have updated the manuscript to explicitly state this to avoid any potential misunderstanding. |
Page 25/ Line 504-507 |
|
There are many grammar problems with spaces and commas, as well as starting the sentences with small letters. Please check your text before uploading. Also, please indicate all authors in the References list, I suggest using Endnote with MDPI style. |
Thank you for your feedback. We have thoroughly reviewed the manuscript to correct grammar issues. Additionally, we have updated the References list to include all authors and formatted it according to the MDPI style using Endnote. |
Page 31 / Line 509-737 |
Reviewer 3 Report
Comments and Suggestions for Authors
Compress the introduction part appropriately;
Thirty-six studies is a little short;
There should also be relevant literature supporting brain-derived neurotrophic factor as a marker.
The table in the paper takes up a lot of space, so the text should be properly supplemented.
Lack of experimental article data;
The format of references should be unified, mainly in the last five years.
Author Response
Response letter
Journal: Biomedicines (ISSN 2227-9059)
Manuscript ID: biomedicines-3320875
Type: Review
Title: Brain-Derived Neurotrophic Factor (BDNF) as a marker of Physical Exercise Effectiveness in Symptom Management:
A Scoping Review
|
Comment |
Response |
Page/Line |
|
Reviewer 3 |
||
|
Compress the introduction part appropriately; |
Thank you for your suggestion. We have revised the introduction to remove redundant content and have emphasized the symptoms included in this review for clarity and focus. |
Page 1-3/ Line 37-123 |
|
Thirty-six studies is a little short; |
Thank you for your observation. We acknowledged that the total number of studies included in this review (n = 35) may be considered limited for the BDNF studies, it is important to note that the inclusion criteria were stringent to ensure that only studies with high relevance to the research question were considered. Given the narrow focus of the review, the number of studies identified is a reflection of the current state of the literature. Despite this, the findings provide valuable insights and underscore the need for further research with larger sample sizes to confirm and expand upon the observed trends. We include in important points in a discussion (Page 24, Line 447-459) Stated that “Our review included only studies conducted on human subjects, which limits the generalizability of the findings to animal models or studies involving non-human organisms. The criteria for inclusion also restricted the studies to those involving healthy volunteers or individuals with physical conditions such as non-communicable chronic conditions (e.g., cancer, neurological disorders, renal disease), excluding studies on mental health conditions (e.g., schizophrenia, bipolar disorder, depression, anxiety, eating disorders) and children under the age of 18. Studies published in languages other than English were also excluded, limiting the review's comprehensiveness and applicability to non-English-speaking populations. Additionally, the review lacks experimental article data, which would have provided stronger evidence for causal relationships. This absence limits the ability to definitively conclude the mechanisms through which exercise impacts BDNF levels and associated symptoms. These factors should be considered when interpreting the findings of this review.” |
Page 24/Line 446-458 |
|
There should also be relevant literature supporting brain-derived neurotrophic factor as a marker. |
Thank you for your comment. We acknowledge the growing interest in BDNF as a potential biomarker and have provided relevant evidence in the introduction (Page 2/ Line 65-76). However, our results indicate that the current findings on BDNF as a marker in human who non-mental health conditions remain limited (Page 21/Line 315-318). We have discussed the role of BDNF as a biomarker for different symptoms in human studies in detail in the discussion section (Page 21-23, Line 325-412). |
Page 2/ Line 65-76 Page 21-23/Line 325-412 |
|
The table in the paper takes up a lot of space, so the text should be properly supplemented. |
Thank you for your comment. To address your concern, we have revised the table to enhance clarity and improve readability. The original table has been separated into two distinct tables: Table 2 (Page 7/ Line 243) now presents study sample characteristics, while Table 3 (Page 10/ Line 259) focuses on interventions, outcomes, and key findings. Additionally, columns related to ELISA kit characteristics and irrelevant details have been removed. This revision reduces the table's size |
Page 7/ Line 243 Page 10/ Line 259 |
|
Lack of experimental article data; |
Thank you for your observation. We acknowledge the impact of lacking experimental article and include statement “Additionally, the review lacks experimental article data, which would have provided stronger evidence for causal relationships. This absence limits the ability to definitively conclude the mechanisms through which exercise impacts BDNF levels and associated symptoms. These factors should be considered when interpreting the findings of this review.” In the limitation section (Page 24/ Line 455-459) |
Page 24/ Line 454-458 |
|
The format of references should be unified, mainly in the last five years. |
Thank you for your comment. Our scoping review aimed to map the breadth of evidence available on the topic, so we did not limit the year of publication during the initial search to ensure comprehensive coverage. However, we have updated the References list to follow the MDPI style using Endnote to ensure consistency and accuracy. |
Page 26-31/ Line 509-737 |
Reviewer 4 Report
Comments and Suggestions for Authors
This review explores the potential role of BDNF as a biomarker in exercise-based intervention for symptoms management. It is expected that it can provide important evidence to suggest the importance of exercise through the BDNF effect through exercise.
The following items are required to be supplemented:
There is a need to supplement the specificity of the title.
What is the basis for dividing exercise types into aerobic, endurance, and relaxation in the abstract?
In the research method, there is a need to organize and present the documents used in the literature search in a table.
Please organize and present Table 1 more systematically.
Please present Table 2 by dividing it into detailed sub-factors. This part is very important. The organization of the results is not systematic at all. Organizing each detailed factor is absolutely required.
Please organize the discussion by dividing it into detailed factors.
Author Response
Response letter
Journal: Biomedicines (ISSN 2227-9059)
Manuscript ID: biomedicines-3320875
Type: Review
Title: Brain-Derived Neurotrophic Factor (BDNF) as a marker of Physical Exercise Effectiveness in Symptom Management:
A Scoping Review
|
Comment |
Response |
Page/Line |
|
Reviewer 4 |
||
|
This review explores the potential role of BDNF as a biomarker in exercise-based intervention for symptoms management. It is expected that it can provide important evidence to suggest the importance of exercise through the BDNF effect through exercise. |
Thank you for your comment. We appreciate your acknowledgment of the review's exploration of BDNF as a biomarker in exercise-based symptom management. |
- |
|
The following items are required to be supplemented: There is a need to supplement the specificity of the title. |
Thank you for your valuable feedback. To address this, we have revised the title, abstract and discussion to specifically highlight the disorders discussed in the paper, namely fatigue, pain, depression, and sleep disturbances. This more focused approach ensures clarity in the paper’s objectives and aligns the discussion with the primary symptoms explored in relation to BDNF and exercise. The revised title now reads: "Brain-Derived Neurotrophic Factor (BDNF) as a Marker of Physical Exercise or Activity Effectiveness in Managing Fatigue, Pain, Depression, and Sleep Disturbances: A Scoping Review." |
Page 1/ Line 2-3 |
|
What is the basis for dividing exercise types into aerobic, endurance, and relaxation in the abstract? |
Thank you for the comment. We classified exercise types based on the American College of Sport Medicine (ACSM) Physical Activity Guidelines for Americans. We include this information in the introduction (Page 2, Line 76-86) and clarify these categories in the data extraction process (Page 4, Line 174-176) |
Page 2, Line 76-86 Page 4, Line 174-176 |
|
In the research method, there is a need to organize and present the documents used in the literature search in a table. |
Thank you for the comment. We reorganized table 2 (Page 7/ Line 243) by including the research method in table 2 and detail intervention and measures in table 3 (Page 10/ Line 259) to improve the clarity. |
Page 7/ Line 243 Page 10/ Line 259 |
|
Please organize and present Table 1 more systematically. |
We have reorganized Table 1 to improve its clarity and systematic presentation. Specifically, we have separated study design and exercise/physical activity types reported in interventional studies for better readability and understanding. |
Page 5-6/ Line 228 |
|
Please present Table 2 by dividing it into detailed sub-factors. This part is very important. The organization of the results is not systematic at all. Organizing each detailed factor is absolutely required. |
Thank you for your feedback. We have revised table 2 (Page 7/ Line 243) to enhance clarity and readability by dividing it into two distinct tables: Table 2 now focuses on study sample characteristics, while Table 3 (Page 10/ Line 259) presents intervention details, outcomes, and key findings, based on study population factors, which are critical in understanding the impact on BDNF changes and symptoms. |
Page 7/ Line 243 Page 10/ Line 259 |
|
Please organize the discussion by dividing it into detailed factors. |
Thank you for your valuable feedback. We have reorganized the discussion section by focusing on the potential of BDNF as a biomarker for symptoms (fatigue, pain, sleep, and depression). Additionally, we have separated the paragraphs discussing BDNF as a mechanistic biomarker and as a mediator in symptom improvement (Page 23/ Line 387 – 401). To further enhance clarity, we have created a new diagram (Figure 2: Exercise/PA-induced BDNF Modulation and its Impact on Fatigue, Pain, Sleep, and Depression) that illustrates how exercise increases BDNF levels and contributes to symptom relief (Page 21/ Line 312-313). |
Page 23/ Line 387–401 Page 21/ Line 312 – 313 |
Reviewer 5 Report
Comments and Suggestions for Authors
According to the manuscript titled "Brain-Derived Neurotrophic Factor (BDNF) as a marker of Physical Exercise Effectiveness in Symptom Management: A Scoping Review". In recent years, there has been considerable interest in establishing the role of Brain-Derived Neurotrophic Factor (BDNF) as a mechanistic marker or therapeutic target for managing symptoms such as depression and cognitive dysfunction as well as improving general well-being. The clinical relevance of BDNF in the management of symptoms remains unclear, however, due to the variability in BDNF response to exercise. Consequently, we conducted a scoping review to assess existing studies that explored the relationship between exercise/physical activity, symptoms, and BDNF levels in adults. Based on the research question, we devised a search strategy to identify relevant studies from 2010 to 2024 indexed in PubMed and CINAHL. Two reviewers evaluated the full-text of the article using Covidence. The data were charted and analyzed in a descriptive manner. A total of 37 records were reviewed out of 768 records. Multiple studies have not found a significant association between self-reported symptoms and changes in peripheral BDNF levels, despite the broader consensus regarding the use of exercise/physical activity to manage symptoms such as fatigue, depression, and cognitive dysfunction. It should be noted, however, that most of the selected studies reported significant increases in BDNF levels following the implementation of an exercise program or an increase in physical activity. A significant difference in BDNF levels and symptoms was not observed between different types of exercise (e.g., aerobic, endurance, and relaxation). In regards to the present manuscript, I would like to make a few comments
-
Perhaps the introduction should discuss the methods or determinations used to determine the BDNF. Also, the range of BDNF values in circulation.
-
In accordance with the PRISMA guidelines, more than two databases should be searched, please verify this
-
The location of the quality values of the studies measured by Joanna Briggs
-
Perhaps the table should be divided by pathology
-
Perhaps the references are more appropriate than the ID in table 3
Author Response
Response letter
Journal: Biomedicines (ISSN 2227-9059)
Manuscript ID: biomedicines-3320875
Type: Review
Title: Brain-Derived Neurotrophic Factor (BDNF) as a marker of Physical Exercise Effectiveness in Symptom Management:
A Scoping Review
|
Comment |
Response |
Page/Line |
|
Reviewer 5 |
||
|
According to the manuscript titled "Brain-Derived Neurotrophic Factor (BDNF) as a marker of Physical Exercise Effectiveness in Symptom Management: A Scoping Review". In recent years, there has been considerable interest in establishing the role of Brain-Derived Neurotrophic Factor (BDNF) as a mechanistic marker or therapeutic target for managing symptoms such as depression and cognitive dysfunction as well as improving general well-being. The clinical relevance of BDNF in the management of symptoms remains unclear, however, due to the variability in BDNF response to exercise. Consequently, we conducted a scoping review to assess existing studies that explored the relationship between exercise/physical activity, symptoms, and BDNF levels in adults. Based on the research question, we devised a search strategy to identify relevant studies from 2010 to 2024 indexed in PubMed and CINAHL. Two reviewers evaluated the full-text of the article using Covidence. The data were charted and analyzed in a descriptive manner. A total of 37 records were reviewed out of 768 records. Multiple studies have not found a significant association between self-reported symptoms and changes in peripheral BDNF levels, despite the broader consensus regarding the use of exercise/physical activity to manage symptoms such as fatigue, depression, and cognitive dysfunction. It should be noted, however, that most of the selected studies reported significant increases in BDNF levels following the implementation of an exercise program or an increase in physical activity. A significant difference in BDNF levels and symptoms was not observed between different types of exercise (e.g., aerobic, endurance, and relaxation). In regards to the present manuscript, I would like to make a few comments |
Thank you for your detailed summary and thoughtful comments on our manuscript. We appreciate your interest in our work and the insights you have provided. We carefully considered your feedback and addressed your comments thoroughly in the revised manuscript. |
- |
|
Perhaps the introduction should discuss the methods or determinations used to determine the BDNF. Also, the range of BDNF values in circulation. |
Thank you for the comment. We include the method and range of circulation BDNF values in the introduction (Page 2/ Line 59-64) |
Page 2/Line 57-64 |
|
In accordance with the PRISMA guidelines, more than two databases should be searched, please verify this |
Thank you for your valuable feedback. In this scoping review, we selected PubMed and CINAHL Plus as the primary databases due to their broad coverage of biomedical and health-related literature, which is central to the scope of our review. We include statement “Only two databases, PubMed and CINAHL Plus, were used as they provide comprehensive coverage of biomedical and healthcare-related studies relevant to our research question. These two databases were selected based on their established relevance in biomedical research and health science. We believe they capture a broad range of relevant articles in our field. Although the inclusion of additional databases could potentially increase coverage, we found these two databases sufficient for the scope of this scoping review.” (Page 4, Line 141-147)
And include “One limitation of this scoping review is the decision to restrict the database search to only PubMed and CINAHL Plus. While these databases provided substantial coverage of the relevant literature in biomedical and healthcare research, using a limited number of databases may have resulted in the omission of relevant studies available in other databases. Expanding the search strategy to include additional databases could have captured a wider range of studies, increasing the comprehensiveness of the review.” (Page 23/ Line 413-418) |
Page 3/ Line 141-147 Page 23, Line 413-418 |
|
The location of the quality values of the studies measured by Joanna Briggs |
We appreciated your comment. According to JBI reviewer’s manual 2015 “a formal assessment of methodological quality of the included studies of a scoping review is generally not performed.” We, however, agree with your comment and include the lack of quality evaluation as one of our limitations in the discussion stated that “It is a strength that a scoping review allows for a broad assessment of the existing literature. However, it does not involve a thorough evaluation of study methodologies, risk of bias, and the strength of the evidence, which allows for more precise conclusions.” (Page 23/Line 420-423) |
Page 24/ Line 420-423 |
|
Perhaps the table should be divided by pathology |
Thank you for your feedback. We have revised Table 2 to enhance clarity and readability by dividing it into two distinct tables: Table 2 (Page 7/ Line 243) now focuses on study sample characteristics, while Table 3 (Page 10/ Line 259) presents intervention details, outcomes, and key findings, based on study population factors, which are critical in understanding the impact on BDNF changes and symptoms. |
Page 7/ Line 243 Page 10/ Line 259 |
|
Perhaps the references are more appropriate than the ID in table 3 |
Thank you for your feedback. We have updated Tables 2–4 to include the last name of the first author, the year of publication, and in-text citations, replacing the IDs to ensure easier and more appropriate access to the references. |
Page 7/ Line 243 Page 10/ Line 259 Page 20/ Line 299 |
Round 2
Reviewer 1 Report
Comments and Suggestions for Authors
I congratulate the authors for making significant modifications to the manuscript, greatly improving it. Therefore, I approve its publication in this journal.
Author Response
Comment 1: I congratulate the authors for making significant modifications to the manuscript, greatly improving it. Therefore, I approve its publication in this journal.
Response 1: Thank you for your kind words and positive feedback. We greatly appreciate your thoughtful review and are pleased to hear that the revisions have improved the manuscript to your satisfaction. Your insights have been invaluable in strengthening our work, and we are grateful for your support in advancing this research for publication.
Reviewer 2 Report
Comments and Suggestions for Authors
Authors have addressed most of my comments and the manuscript seems to be less misleading in the parts described in my previous reports.
However, several minor remarks are introduced within the new text (such as absent/not straight lines in Fig.2 and lack of “4.4” and “4.5” within the corresponding paragraph titles), plus native English speaker should check new text parts (starting with “full text based” in line 23, which probably should be “full texts, based”).
Still no understanding how Covidence platform is used in your search, does it calculate the percentages for you then? Or simply extract the useful information from full-text? Or is it just used as a platform for co-working? Because, it now seems you do not to mention this at all, or if somehow you have to, please explain, which of its functions help you in writing the manuscript.
And please move Table 3 to Supplementary, it’s 1/3 of the manuscript and сhokes down the search of more useful information!
Comments on the Quality of English Language
Described in the main section of the Comments.
Author Response
Comment 1: Authors have addressed most of my comments and the manuscript seems to be less misleading in the parts described in my previous reports.
Response 1: Thank you for your thoughtful review and for acknowledging our efforts to address your comments. We are pleased to hear that the revised manuscript is now clearer and aligns better with your feedback.
Comment 2: However, several minor remarks are introduced within the new text (such as absent/not straight lines in Fig.2 and lack of “4.4” and “4.5” within the corresponding paragraph titles), plus native English speaker should check new text parts (starting with “full text based” in line 23, which probably should be “full texts, based”).
Response 2: Thank you for your careful review and for pointing out these minor issues. We have addressed the concerns as follows:
- The figure lines in Figure 2 have been corrected to ensure they are straight and properly aligned (page 11, Line 322)
- The missing section numbers "4.4" (page 13, Line 424) and "4.5" (page 15, Line 481) have been added to the corresponding paragraph titles for consistency and clarity.
- The suggested phrase in line 23 has been revised to “full texts, based” as recommended, and a native English speaker has reviewed the entire revised text to improve readability and grammatical accuracy. The changes have been highlighted in red based on the editorial suggestions.
Comment 3: Still no understanding how Covidence platform is used in your search, does it calculate the percentages for you then? Or simply extract the useful information from full-text? Or is it just used as a platform for co-working? Because, it now seems you do not to mention this at all, or if somehow you have to, please explain, which of its functions help you in writing the manuscript.
Response 3: Thank you for your valuable feedback. We apologize for any confusion regarding the role of the Covidence platform in our systematic review process. Covidence was primarily used to facilitate the organization and management of the review workflow, specifically in study selection, data extraction, and reviewer collaboration. It was a management tool to ensure consistency, enabling efficient co-working and progress tracking. Covidence does not have the capability to calculate percentages or automatically extract data from full texts. Instead, it provided a structured environment where reviewers could independently screen titles and abstracts, perform full-text reviews, and extract relevant data using pre-defined templates. Discrepancies identified during these stages were resolved through discussions or adjudication by a third reviewer. We have now included a detailed explanation of Covidence's role in the revised method section, page 4, Lines 151-155, Lines 157-160, and Lines 194-197 to clarify how it supported this review's development.
Comment 4: And please move Table 3 to Supplementary, it’s 1/3 of the manuscript and сhokes down the search of more useful information!
Response 4 Thank you for your feedback. We acknowledge your concern regarding the length of Table 3 and its impact on the manuscript's readability. As suggested, we have moved Table 3 to the Supplementary Materials section and updated the main text to reflect this change (page 9, lines 258-259) to enhance the flow of the main text and facilitate easier access to key information.
We sincerely appreciate your constructive feedback and your dedicated time to review our manuscript. Your valuable insights have significantly contributed to improving our work's clarity, organization, and overall quality. We believe the revisions made in response to your comments have strengthened the manuscript, and we hope it now meets the journal's expectations.
Reviewer 4 Report
Comments and Suggestions for Authors
It was well revised as review. You did a good job.
Author Response
Point-by-point response to comments and suggestions for authors
Comment 1: It was well revised as review. You did a good job.
Response 1: Thank you for your kind words and positive feedback. I greatly appreciate your time and effort in reviewing the revised manuscript.